# Mechanism and Performance of the SCR of NO with NH₃ over Sulfated Sintered Ore Catalyst

**Wangsheng Chen** [1,2]**, Fali Hu** [1,2]**, Linbo Qin** [1,2]**, Jun Han** [1,2,*] **, Bo Zhao** [1,2]**, Yangzhe Tu** [1,2] **and Fei Yu** [3]

[1] School of Resource and Environmental Engineering, Wuhan University of Science and Technology, Wuhan 430081, China; chenwangsheng@wust.edu.cn (W.C.); hufali@hotmail.com (F.H.); qinlinbo@wust.edu.cn (L.Q.); zhaobo87@wust.edu.cn (B.Z.); tuyangzhe123@hotmail.com (Y.T.)

[2] Industrial Safety Engineering Technology Research Center of Hubei Province, Wuhan University of Science and Technology, Wuhan 430081, China

[3] Department of Agricultural and Biological Engineering, Mississippi State University, Mississippi State, MS 39762, USA; fyu@abe.msstate.edu

* Correspondence: hanjun@wust.edu.cn; Tel.: +86-27-6886-2880

**Abstract:** A sulfated sintered ore catalyst (SSOC) was prepared to improve the denitration performance of the sintered ore catalyst (SOC). The catalysts were characterized by X-ray Fluorescence Spectrometry (XRF), Brunauer–Emmett–Teller (BET) analyzer, X-ray diffraction (XRD), X-ray photoelectron spectroscopy (XPS) and diffuse reflectance infrared spectroscopy (DRIFTS) to understand the NH₃-selective catalytic reduction (SCR) reaction mechanism. Moreover, the denitration performance and stability of SSOC were also investigated. The experimental results indicated that there were more Brønsted acid sites at the surface of SSOC after the treatment by sulfuric acid, which lead to the enhancement of the adsorption capacity of NH₃ and NO. Meanwhile, Lewis acid sites were also observed at the SSOC surface. The reaction between $-NH_2$, $NH_4^+$ and NO (E-R mechanism) and the reaction of the coordinated ammonia with the adsorbed $NO_2$ (L-H mechanism) were attributed to $NO_x$ reduction. The maximum denitration efficiency over the SSOC, which was about 92%, occurred at 300 °C, with a 1.0 NH₃/NO ratio, and 5000 h⁻¹ gas hourly space velocity (GHSV).

**Keywords:** sintered ore catalyst; sulfate; In-situ DRIFTS; SCR

---

## 1. Introduction

Nitrogen oxide ($NO_x$) is one of the major atmospheric pollutants and is mainly generated from the combustion of fossil fuel, which has serious harmful effects on human health and the ecological environment. In 2016, the total emission of NOx from the iron and steel industry was about 1.04 million tonnes [1]. The sintering process in iron-making plants uses coal or coke as fuel, which is a major emission source of NOx. About 35–50% of the total NOx emission from the iron and steel industry is attributed to the sintering process [2–4]. The new ultra-low emission standard of air pollutants for the iron and steel industry will be issued by Chinese government and require that NOx concentration in the sintering flue gas should be below 50 mg/m³. Hence, it is urgent to address the treatment of sintering flue gas.

Selective catalytic reduction (SCR) over $V_2O_5$-$WO_3$ ($MoO_3$)/$TiO_2$ catalyst has been widely applied in power stations because of its high denitration efficiency [5–7]. However, there are still shortcomings such as the toxicity of the catalyst, and the high operation costs [8]. In particular, the reaction temperature of the current commercial catalysts is higher than the temperature of the sintering flue gas [9]. Therefore, the sintering flue gas must be heated to the reaction temperature of the catalyst by using additional fuels.

　　Many researchers paid more attention to developing the low temperature catalysts or the catalysts prepared by the non-noble metals for sintering flue gas. Zhang et al. reported that the Fe–based catalysts exhibited a high catalytic activity for NO reduction [10]. Wang [11] also stated that the $Fe_2O_3$ particles had a good performance during $NO_x$ elimination, where the highest $NO_x$ conversion reached 95%. Yang et al. [12] claimed that $\alpha$–$Fe_2O_3$ had a poor SCR activity, while $\gamma$–$Fe_2O_3$ had an excellent SCR activity at 200–350 °C. Meanwhile, it was also found that the further increase reaction temperature from 350 to 500 °C would suppress NOx conversion over $\gamma$–$Fe_2O_3$ [13].

　　It was well known that the sintered ore was one of the raw materials in the sintering plant, its main component was $Fe_2O_3$. In our previous study, Han et al. and Chen et al. [14,15] proposed that the hot sintered ore was used as catalysts for $NO_x$ removal from the sintering flue gas. The experimental results illustrated that the hot sintered ore had a good denitration performance, the denitration efficiency was about 60% at 300 °C, with a 1.0 $NH_3$/NO ratio, and 1000 h$^{-1}$ GHSV. However, the denitration efficiency was too low and the NOx concentration from the sintering flue gas could reach the limit after SCR over the sintered ore.

　　Ciambelli [16] found that introduction of $SO_4^{2-}$ into SCR catalysts can promote the surface acidity of the catalysts. Thus, the catalytic activity was promoted. Khodayari [17] and Xu [18] thought that the sulfation of the catalysts would decrease the oxidization ability of $Fe^{3+}$, and separated the active sites of adsorbing –$NH_2$ and the active sites of oxidizing –$NH_2$. Therefore, the catalytic oxidization of $NH_3$ over $\gamma$–$Fe_2O_3$ was suppressed. This process resulted in an obvious promotion of NOx conversion. Zhang [19] also investigated the sulfation of $CeO_2$-$ZrO_2$, and their experiment results demonstrated that the sulfated $CeO_2$-$ZrO_2$ provided more surface acidities and acidic sites, and Brønsted acid sites were also increased.

　　In order to improve the denitration efficiency, SSOC was prepared and the influence of the acidification on SCR performance was investigated. At the same time, the reaction mechanisms of SCR over SSOC were also discussed.

## 2. Results and Discussion

### 2.1. Characterizations

　　The main components of the two catalysts were shown in Table 1. It presented that the main components of both two catalysts were $Fe_2O_3$ and CaO. After the introduction of sulfuric acid, the proportion of iron oxide in the catalyst was decreased and sulphate content was significantly increased.

**Table 1.** Chemical compositions of catalysts (%).

|  | $Fe_2O_3$ | CaO | $SiO_2$ | $Al_2O_3$ | MgO | $MnO_2$ | $TiO_2$ | $P_2O_5$ | $SO_x$ |
|---|---|---|---|---|---|---|---|---|---|
| SOC | 75.42 | 12.54 | 4.22 | 1.19 | 1.02 | 0.19 | 0.20 | 0.09 | 0.14 |
| SSOC | 48.92 | 11.52 | 2.82 | 0.79 | 0.85 | 0.13 | 0.09 | 0.05 | 29.92 |

　　The BET surface area, pore-size and pore volume of the catalysts were presented in Table 2. The specific surface area of the SOC was 3.684 m$^2$/g, the total pore volume was 0.00714 cm$^3$/g, and the average pore diameter was 6.3 nm. The BET surface area, pore size and pore volume of the SSOC were 4.734 m$^2$/g, 5.9 nm and 0.00964 cm$^3$/g, respectively. It was indicated that the introduction of sulfuric acid would open some micro-pores, which lead to the increase of the total pore volume. At the same time, the sulphate would block some pores. The BET of SSOC was higher than that of SCO, which meant that the sulfation had a positive effect on the pore structure of SOC.

**Table 2.** Textural properties of the catalysts.

| Catalysts | BET Surface Area ($m^2 \cdot g^{-1}$) | Total Pore Volume ($cm^3 \cdot g^{-1}$) | Average Pore Diameter (nm) |
|:---:|:---:|:---:|:---:|
| SOC | 3.684 | 0.00714 | 6.3 |
| SSOC | 4.734 | 0.00964 | 5.9 |

The XRD patterns of the catalysts were demonstrated in Figure 1. Several sharp diffraction peaks at 24.1°, 33.2°, 35.6°, 49.6°, 54.2°, 57.1° and 62.6° were observed, which were assigned to $\alpha$-$Fe_2O_3$ (JCPDS NO. 33-0664), and the characteristic peaks of $\alpha$-$Fe_3O_4$ at 30.1°, 33.2°, 49.6°, 54.2° and 57.1° were also detected in Figure 1. The results indicated that the main components of SOC were $\alpha$-$Fe_2O_3$ and a small amount of $\alpha$-$Fe_3O_4$. After the acidification with sulfuric acid, it could be seen that the peak of $\alpha$-$Fe_2O_3$ was decreased. Meanwhile, the characteristic peaks (25.4°, 31.4°, 38.7°, 52.2°) of $Fe_2(SO_4)_3$ were detected, and a small amount of $FeSO_4$ and $CaSO_4$ also appeared.

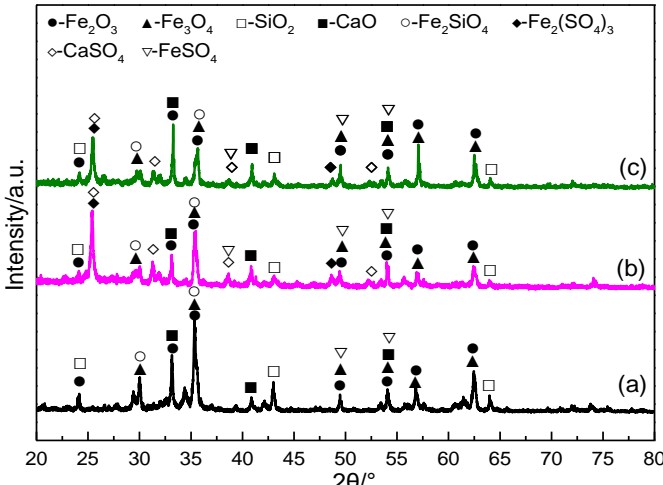

**Figure 1.** XRD patterns of the catalysts: SOC (**a**), SSOC (**b**) and SSOC after tested (**c**).

X-ray photoelectron spectroscopy (XPS) was used to study the valence states of the catalysts, as shown in Figure 2. It could be seen from Figure 2 (I) that there were obvious peaks of C (1s), O (1s), and Fe (2p) in both of the two catalysts. Moreover, there were obvious S (2p) peaks after the introduction of sulfuric acid. As shown in Figure 2 (II), the binding energies of Fe 2p 2/3 and Fe 2p 1/2 on SOC were mainly centered at about 710.8 and 724.3 eV, which were indications that the iron species were mainly in the form of $Fe^{3+}$ in the SOC [20]. At the same time, a small number of $Fe^{2+}$ existed, which was also consistent with the XRD results. The binding energies of Fe 2p 2/3 and Fe 2p 1/2 of SSOC were higher than those of SOC. The peak of Fe 2p 2/3 appeared at 711.8 eV and the peak of Fe 2p 1/2 appeared at 724.9 eV. After the acidification with sulfuric acid, both $Fe_2(SO_4)_3$ and $FeSO_4$ appeared in the SSOC, which contributed to Fe 2p peaks shifting to the higher position. In addition, in Figure 2 (III), the peak of S 2p appeared at 169.3 eV, indicating that sulfur compounds existed in the form of $SO_4^{2-}$ [21].

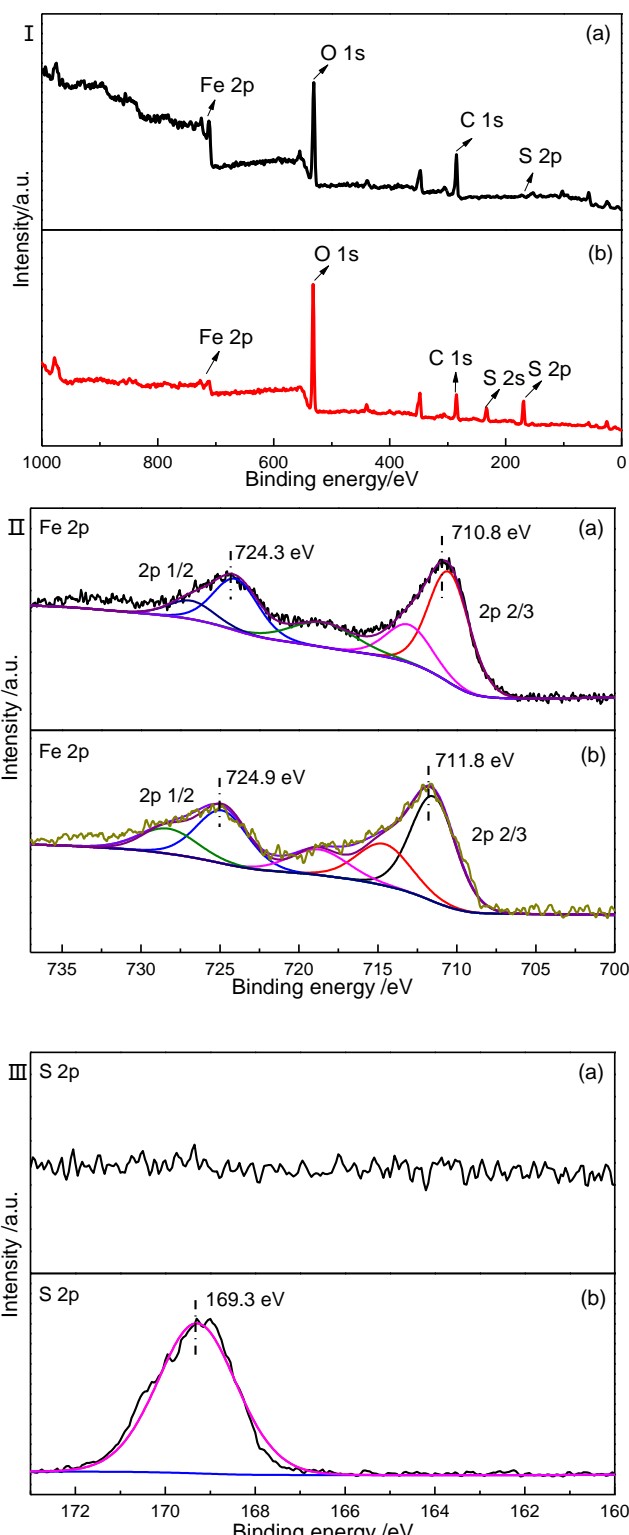

**Figure 2.** XPS spectra of the catalysts: SOC (**a**) and SSOC (**b**).

## 2.2. In-Situ DRIFTS Studies

### 2.2.1. Adsorption of NH$_3$

Figure 3 showed the FTIR spectra of NH$_3$ adsorption over the SSOC with 1000 ppm NH$_3$/Ar under different temperatures (250–350 °C). Thirty minutes after NH$_3$/Ar introduction, the bands at

1690, 1525, 1460 and 1408 cm$^{-1}$ were observed. The bands at 1690, 1460 and 1408 cm$^{-1}$ were symmetric bending vibration of NH$_4^+$ species at Brønsted acid sites [20,22]. The bands at 1525 and 3434 cm$^{-1}$ were assigned to the intermediate species, such as ammonium or amide species (–NH$_2$) [20]. Meanwhile, the bands at, 3242 and 3120 cm$^{-1}$ could be ascribed to the N–H stretching vibration of coordination NH$_3$, and the bands at 1259 cm$^{-1}$ (symmetric bending vibration of NH$_3$ on Lewis acid sites) and 1102 cm$^{-1}$ (NH$_3$ species adsorbed on Lewis acid sites) also appeared [20,23,24]. The intensities of the peaks of NH$_4^+$ species (1690, 1460 and 1408 cm$^{-1}$) and –NH$_2$ species (1525 cm$^{-1}$) first increased and then decreased in the temperature range of 250–350 °C. There were peaks at 960 and 930 cm$^{-1}$ (NH$_3$ of gaseous state or weak adsorption state) under 300 °C. Yu et al. [25] stated that the following reactions probably took place in the reaction:

$$NH_3(g) \rightarrow NH_3(a) \tag{1}$$

$$NH_3(a) + O(a)/O^{2-} \rightarrow NH_2(a) + OH(a) \tag{2}$$

$$NH_3(g) + H^+ \rightarrow NH_4^+ \tag{3}$$

According to Equations (1)–(3), the more functional groups formed at the surface of the catalyst, the higher the reaction activity. Thus, the optimum adsorption activity of NH$_3$ over the SSOC was 300 °C, as shown in Figure 3.

The dependence of FTIR spectra over the SOC and SSOC on the reaction time at 1000 ppm NH$_3$/Ar and 300 °C was presented in Figure 4. Figure 4a demonstrated that there were 6 bands in the range of 1690–1100 cm$^{-1}$. The bands at 1690, 1405 and 1454 cm$^{-1}$ were related to the symmetric bending vibration of NH$_4^+$ species, and the band at 1525 cm$^{-1}$ was ascribed to the intermediate products of ammonium or –NH$_2$ species. Moreover, the band at 3242 cm$^{-1}$ (N–H stretching vibration of coordinated NH$_3$) appeared at 10 min. With the feed of NH$_3$, the adsorption band at 1259 cm$^{-1}$ (symmetric bending vibration of NH$_3$ on Lewis acid sites) appeared at 30 min, and the band at 1107 cm$^{-1}$ (NH$_3$ of gaseous state or weak adsorption state) appeared at 60 min. Hence, there were both Lewis acid sites and Brønsted acid sites on the surface of SOC. The FTIR spectra over SSOC dependence of the reaction time was presented in Figure 4b, and the bands of NH$_3$ adsorption were basically the same as those of SOC. Moreover, the bands at 3434 cm$^{-1}$ (–NH$_2$ groups) and 3128 cm$^{-1}$ (coordination NH$_3$ on Lewis acid sites) were observed because the surface acidity of SSOC was strengthened. There were two new weak bands that appeared at 965 and 927 cm$^{-1}$, where NH$_3$ was in a gaseous state or weak adsorption state. These weakly adsorbed or gaseous NH$_3$ could rapidly adsorbed on the acid sites once the adsorbed NH$_3$ species was consumed. It could also be seen that the strength of NH$_3$ adsorption peak was enhanced after SOC acidification with sulfuric acid, which was the reason that the surface acidity of SSOC was enhanced after the acidification. The influence of the acidity of SSCO on the dentiration efficiency is presented in Figure 5. The different acidity of SSOC was achieved by sulfating with 1, 3 and 5 mol/L sulfuric acid. In this experimental run, the mass of catalyst and the volume of sulfuric acid were the same. Only the concentration of sulfuric acid was varied. It was demonstrated the maximum denitration efficiency occurred at the catalyst treated by 5 mol/L sulfuric acid, and its denitration efficiency was 92.3% at 300 °C. At the same time, the denitration efficiency of the catalysts sulfated by 1 and 3 mol/L was 56.6% and 68.5% at the same reaction temperature. Moreover, the experimental results proved that the optimum reaction temperature for all catalysts was 300 °C. These sulfates on the surface of SSOC provided more Brønsted acid sites (Peaks at 1690, 1525 cm$^{-1}$ were increased), which promoted the adsorption capacity for NH$_3$. Hence, the catalyst denitration performance was also improved [26,27].

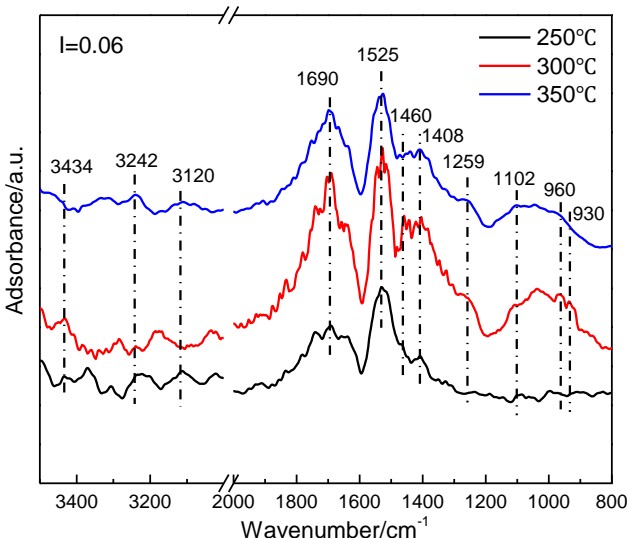

**Figure 3.** DRIFT spectra of SSOC in the condition of 1000 ppm NH$_3$ at 250–350 °C for 30 min.

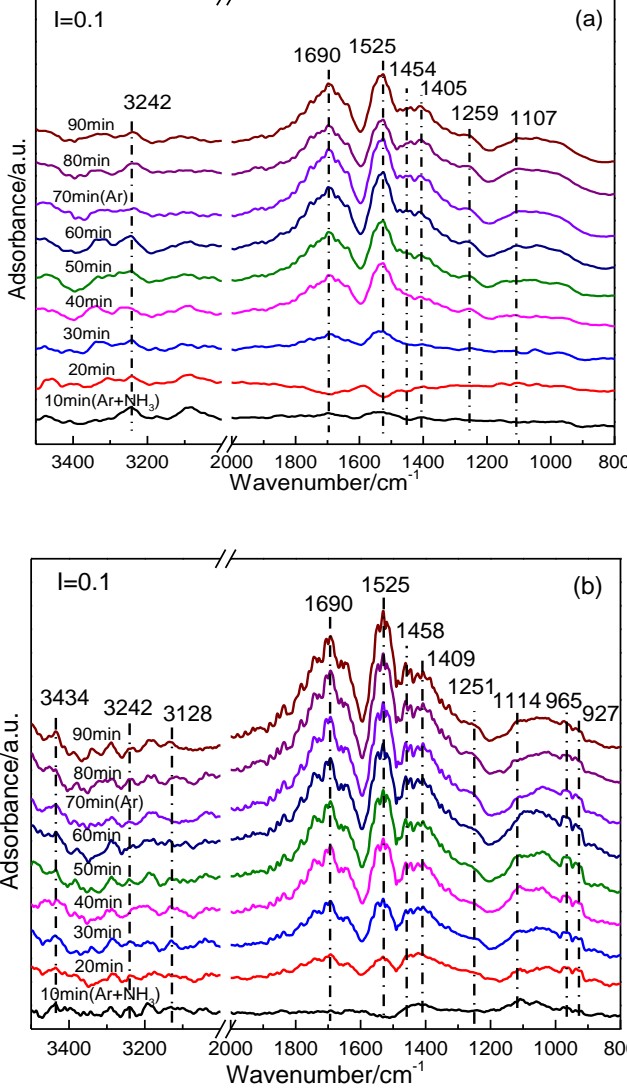

**Figure 4.** DRIFT spectra of SOC (**a**) and SSOC (**b**) exposed to 1000 ppm NH$_3$ at 300 °C at different time.

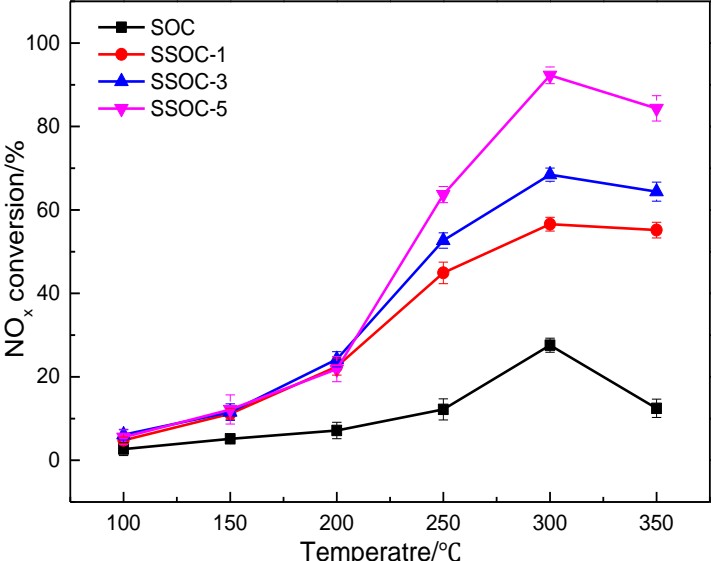

**Figure 5.** The influence of acidity of SSOC on SCR performance.

### 2.2.2. Co-Adsorption of NO and $O_2$

Figure 6 indicated the DRIFTS spectra of co–adsorption of NO and $O_2$ under 250–350 °C, and 1000 ppm NO + 15% $O_2$/Ar. After 30 min, there were 4 bands appeared in the range of 1620–1290 $cm^{-1}$ and a weak band appeared at 1014 $cm^{-1}$. The band at 1607 $cm^{-1}$ was ascribed to bridged nitrate species and adsorbed $NO_2$ molecules [28,29]. The bands at 1485 and 1417 $cm^{-1}$ were assigned to bidentate nitrates and monodentate nitrates respectively [30–32]. In addition, the bands at 1293 and 1014 $cm^{-1}$ were related to nitro compounds [31]. The variation trend of the intensities of these bands were same as $NH_3$ adsorption in the temperature range of 250–350 °C. The adsorption process can be explained by the following formula:

$$NO(g)NO + O(a) \leftrightarrow NO_2(a) - Bridgednitrites \tag{4}$$

$$NO(g) + O(a) \leftrightarrow NO_2(a) - Monodentatednnitrites \tag{5}$$

$$NO_2(g) + O(a) \leftrightarrow NO_3(a) - Bidentatenitrites \tag{6}$$

$$NH_2(a) + NO(g) \rightarrow NH_2NO \rightarrow N_2 + H_2O \tag{7}$$

$$NH_4^+(a) + NO(g) \rightarrow \{NH_3NO \rightarrow NH_2NO + H_2O\} \rightarrow N_2 + H_2O \tag{8}$$

$$NO_2(a) + NH_3(a) \rightarrow NO_2[NH_3]_2(a) + NO \rightarrow 2N_2 + 3H_2O \tag{9}$$

$$NO_2(a) + 2NH_4^+(a) \rightarrow NO_2NH_4^+(a) + NO \rightarrow N_2 + 2H_2 \tag{10}$$

According to Equations (4)–(10), all the peaks in Figure 6 were important to the SCR reaction. The intensity of peaks at 300 °C was the highest, which mean that the optimum adsorption temperature for NO was also 300 °C.

Figure 7 showed the DRIFTS spectra over SOC and SSOC at different reaction times under 300 °C, 1000 ppm NO + 15% $O_2$/Ar. Similar with the spectra in Figure 6, after 10 min, there were 4 bands that appeared in the range of 1620–1300 $cm^{-1}$ and a weak band appeared at 1022 $cm^{-1}$. The bands intensities were gradually increased with the adsorption time. It could be seen that in Figure 7b, the adsorption intensity of SSOC was obviously higher than that of SOC, especially for the nitro compounds (1290 $cm^{-1}$) and the nitrate species (1490 $cm^{-1}$). The results indicated that the adsorption capacity of NO was improved after the acidification with sulfuric acid. It had been shown that the introduction of $SO_4^{2-}$ enhanced the adsorption of NO [26].

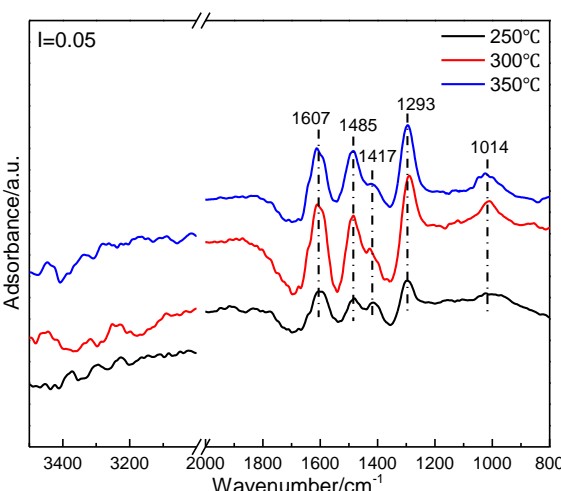

**Figure 6.** DRIFT spectra of SSOC in the condition of 1000 ppm NO and 15% $O_2$ at 250–350 °C for 30 min.

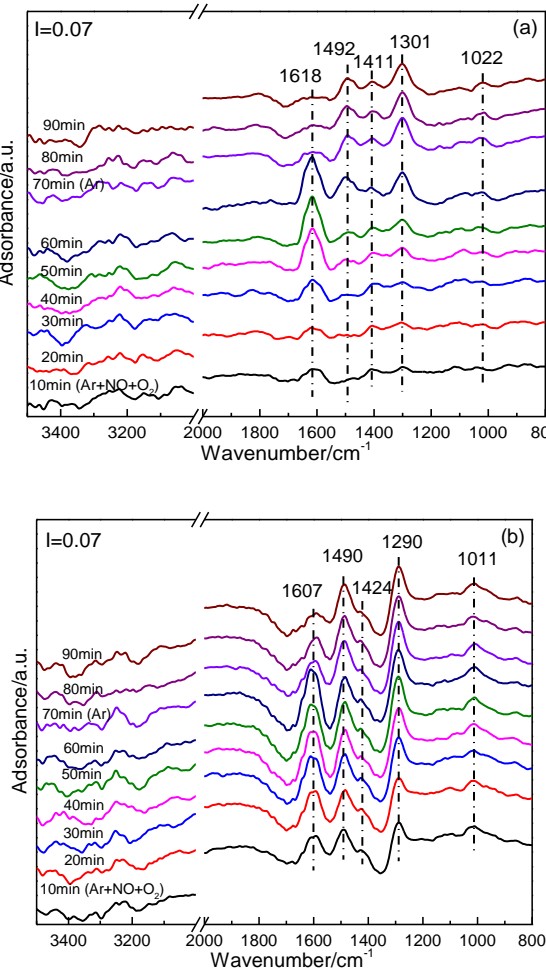

**Figure 7.** DRIFT spectra of SOC (**a**) and SSOC (**b**) exposed to 1000 ppm NO and 15% $O_2$ at 300 °C for a different time.

The results showed that the peak intensities of the adsorbed species related to $NO_2$ molecules and bridged nitrate species (1607 and 1618 $cm^{-1}$) were decreased obviously with purging by Ar, which indicated $NO_2$ molecules and bridged nitrate species absorbed at the surface of catalysts were unstable.

At the same time, it was found that the intensities of bidentate nitrates, monodentate nitrates and nitro compounds were stable even after an Ar purge.

### 2.2.3. Reaction between NH$_3$ and NO

The reaction at the surface of catalyst by introducing NH$_3$ and NO + O$_2$ into an in-situ reactor was also studied. Figure 8 shows that NH$_4^+$ species (1695 cm$^{-1}$), amide species(–NH$_2$) or ammonium (1533 cm$^{-1}$), and several weak adsorption bands at 1252, and 3242 cm$^{-1}$ (NH$_3$ adsorbed on Lewis acid site) were detected on the surface of SOC after NH$_3$ introduction at 30 min. On the surface of SSOC catalyst, NH$_4^+$ species at 1695, 1427 and 1454 cm$^{-1}$ and weakly adsorbed NH$_3$ or gaseous NH$_3$ (966, 925 cm$^{-1}$) were also detected. There were two bands at 3434 cm$^{-1}$ (–NH$_2$ groups) and 3127 cm$^{-1}$ (coordination NH$_3$ on Lewis acid sites) that appeared at the same time. After switching to NO + O$_2$, the adsorbed species of NH$_3$ over SOC and SSOC gradually disappeared and the adsorbed species of NO$_x$ appeared. These bands were ascribed to NO$_2$ molecules and bridged nitrate species (1602 and 1618 cm$^{-1}$), bidentate nitrates (1488 and 1498), monodentate nitrates (1413 and 1419 cm$^{-1}$) and nitro compounds (1290, 1295, 1020 and 1011 cm$^{-1}$). Comparing Figure 8a with Figure 8b, it could be seen that there were more nitrite species and NH$_3$ species adsorbed at SSOC than those at SOC. This could be explained that the SSOC contained more active sites, which resulted from the sulfates on SSOC, and Equations (7)–(10) occurred [33]. The reaction of amide species (–NH$_2$) and NH$_4^+$ species on the surface of catalysts with the gaseous NO was E-R mechanism, and the reaction between adsorbed state NO$_2$ and adsorbed NH$_3$ followed L-H mechanism. Therefore, the reaction between NO and NH$_3$ over SOC and SSOC had two mechanisms.

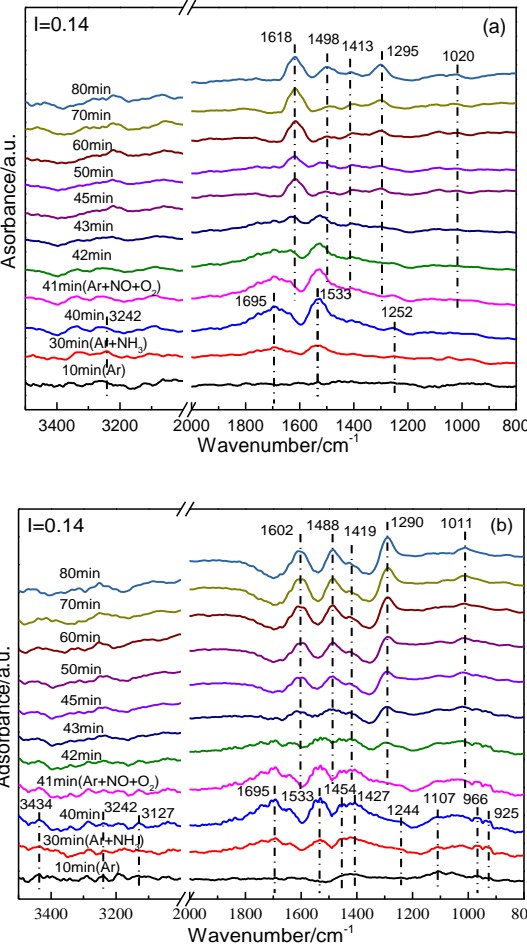

**Figure 8.** DRIFT spectra of SOC (**a**) and SSOC (**b**) successively exposed to 1000 ppm NH$_3$, 1000 ppm NO + 15% O$_2$ for different time under 300 °C.

### 2.2.4. SCR Performance

SCR performance were carried out in a fixed-bed reactor, which was made of quartz with an inner diameter of 20 mm and a length of 1000 mm. In this experiment, the mass of the catalysts samples was 13.49 g, the flow rate of the flue gas which simulated sintering flue gas was about 600 mL/min, the simulated sintering flue gas contained 300 ppm NO, 15% $O_2$ and balance $N_2$. The temperature in the reactor was kept at 100 °C–350 °C, with the condition of a 0.5–1.0 $NH_3$/NO ratio and 5000 $h^{-1}$ GHSV. The $NO_x$ concentrations in simulated flue gas at the inlet and outlet of the reactor were continuously recorded by a gas analyzer (PG-350, Horiba, Kyoto, Japan) with an accuracy of ±1.0%. The $NO_x$ conversion was calculated according to the following equations:

$$NO_x \text{ conversion} = \frac{[NO_x]_{in} - [NO_x]_{out}}{[NO_x]_{in}} \times 100\% \tag{11}$$

Figure 9 presented the denitration performance of SOC and SSOC in the temperature range of 100 °C–350 °C. It was found that the reaction temperature had a great effect on the $NH_3$-SCR denitration performance of SOC and SSOC. The optimum reaction temperature was 300 °C, which conformed well with the in-situ DRIFTS results. The $NO_x$ conversion of SOC was only 27% at 300 °C, 1.0 $NH_3$/NO ratio and 5000 $h^{-1}$ GHSV, and the $NO_x$ conversion of SSOC was 92% at the same condition. It was found that the denitration performance of SSOC was greatly improved and the optimum reaction temperature was 300 °C.

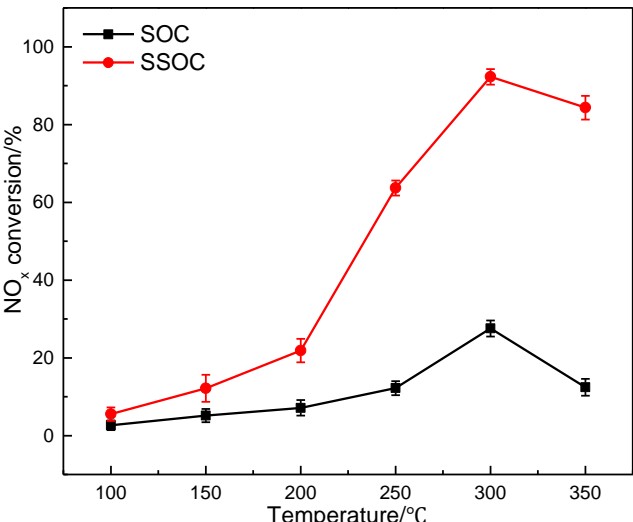

**Figure 9.** Effect of temperature on SCR activity on SOC and SSOC. 300 ppm NO, 15% $O_2$, Ar as balance gas, $NH_3$/NO ratio: 1.0, GHSV: 5000 $h^{-1}$.

The stability test of SSOC was shown at Figure 10. The reaction continued for 24 h at 300 °C, 1.0 $NH_3$/NO ratio, GHSV = 5000 $h^{-1}$. It could be seen that the $NO_x$ conversion was stable at about 92%, which indicated that SSOC has a good denitration stability. Compared with the SOC, the denitration performance had been greatly improved after the acidification with sulfuric acid. The adsorption of $NH_3$ and NO on SSOC was obviously improved, which subsequently promoted the $NH_3$-SCR reaction.

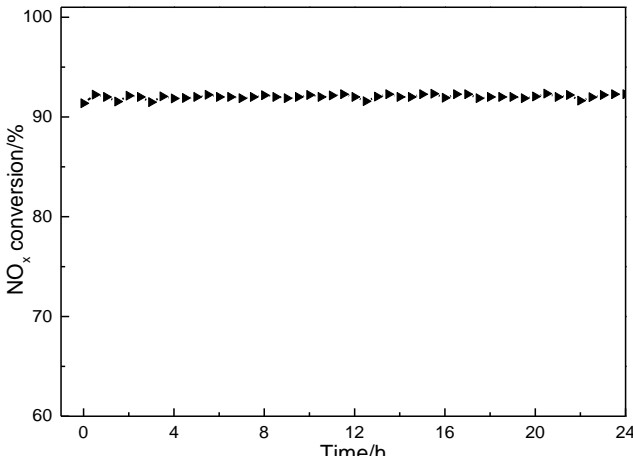

**Figure 10.** The stability test of SSOC. 300 ppm NO, 15% $O_2$, Ar as balance gas, $NH_3$/NO ratio: 1.0, GHSV: 5000 $h^{-1}$.

## 3. Experimental Process

### 3.1. Catalyst Preparation

In this experiment, the sintered ore was sampled from a sintering workshop in Wuhan. The sintered ore was dried, milled and sieved to 0.15–0.25 mm, which was denoted as SOC. The SSOC was prepared by using an impregnation method. Firstly, 100 g sintered ore (0.15–0.25 mm) was weighed and put into a beaker, then 50 mL sulfuric acid solution with a concentration of 5 mol/L was added and stirred simultaneously for 30 min. After filtration and washing with the deionized water, the mixture was dried at 105 °C and then calcined at 500 °C for 3 h in the air atmosphere. Finally, the prepared SSOC were naturally cooled to the room temperature, then crushed and sieved to 0.15–0.25 mm.

### 3.2. Catalyst Characterization

The main chemical composition of the SOC and SSOC were analyzed by X-ray fluorescence spectroscopy (XRF) (ARL SMS-XY, Thermo Fisher Scientific Corp., Waltham, MA, USA). The specific surface area, pore volume and pore size distribution of the catalysts were measured by an automated adsorption analyzer (Micromeritics ASAP 2020, Micromeritics Corp., Norcross, GA, USA). The catalysts samples were firstly degassed at 240 °C for 4 h before the test, and the adsorption medium was liquid nitrogen. The nitrogen adsorption and desorption were analyzed at −196 °C using BET analyzer (Micromeritics ASAP 2020, Micromeritics Corp., Norcross, GA, USA). The surface area was calculated by using the BET method according to nitrogen adsorption data in the relative pressure (P/P0) range of 0.01–1. X-ray diffraction (XRD; Rigaku RINT2000, Tokyo, Japan) was performed using CuKα radiation (λ = 1.54056 Å) to detect the crystalline phases of the samples. The analysis of XRD was referred to International Centre for Diffraction Data (ICDD). X-ray photoelectron spectroscopy (XPS, AXIS Ultra DLD, Shimazu Corp., Kyoto, Japan) was used to determine the valence states of the surface atoms of the catalysts with Al Kα radiation.

In-situ DRIFTS experiments were carried out in a FTIR spectrometer (FT-IR, Bruker Tensor II, Bruker optics Corp., Karlsruhe, Germany) equipped with an in-situ cell and a mercury-cadmium-telluride detector [34–37]. For the adsorption of $NH_3$ (or NO + $O_2$), the catalyst was exposed to a 20 mL/min $NH_3$ (or NO + $O_2$), which resulted in the variation with adsorption time of the DRIFT spectra, and argon purging was subsequently performed. In the reaction mechanism studies, the catalyst was pretreated in a flow of 20 mL/min $NH_3$ for 40 min, then was shifted NO + $O_2$ at 300 °C to get the DRIFT spectra. All spectra were recorded by accumulating 100 scans at a spectra resolution of 4 $cm^{-1}$.

## 4. Conclusions

In this paper, SSOC displayed an excellent catalytic denitration activity. There were some sulfates that appeared in SSOC after acidification with sulfuric acid solution, which provided more Brønsted acid sites. In-situ DRIFTS results demonstrated that there were Lewis and Brønsted acid sites simultaneously on the surfaces of SOC and SSOC. The acidification contributed to the increase of the Brønsted acid sites at SSOC, which improved the adsorption capacity of $NH_3$ and NO. $NH_3$ and NO were adsorbed on the surface of catalysts to form amide species ($-NH_2$), $NH_4^+$ species, $NO_2$ molecules in gaseous or weakly adsorbed state and nitrate species. Meanwhile, it could be seen that the $NH_3$-SCR process of SOC and SSOC followed E-R and L-H mechanisms. Moreover, the optimum reaction temperature of catalysts was 300 °C and maximum $NO_x$ conversion fraction over SSOC was 92% at 1.0 $NH_3$/NO ratio and 5000 $h^{-1}$ GHSV. It was observed that the $NO_x$ conversion could be steadily maintained.

**Author Contributions:** Data curation, F.H.; Investigation, Y.T.; Methodology, L.Q.; Project administration, J.H.; Software, B.Z.; Writing—original draft, W.C.; Writing—review & editing, F.Y.

**Funding:** The present work was partly supported by the National Natural Science Foundation of China (Grant No. 51476118 and 51576146).

**Conflicts of Interest:** The authors declare no conflict of interest. The funders had no role in the design of the study; in the collection, analyses, or interpretation of data; in the writing of the manuscript, or in the decision to publish the results.

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
