# Peer review of "Mechanism and Performance of the SCR of NO with NH3 over Sulfated Sintered Ore Catalyst"

_catalysts, doi:10.3390/catal9010090_

Round 1

Reviewer 1 Report

This manuscript describes interesting results how the catalytic efficiency of sintered ore catalyst for NH3-SCR reaction can be enhanced by using sulfuric acid treatment. However, the authors need to add more specific discussion. Therefore, it needs to address these issues before it can be accepted for publication.

Below is my concerns about this manuscript.

In the DRIFT spectra results of Figure 3, the authors described that the two new peaks (965 and 927 cm-1) enhance the NH3 adsorption strength by sulfuric acid treatment and these sulfates on the surface of SSOC provide more Brønsted acid sites which can improve catalyst performance (p5, line 140). But there is a lack of explanation for the association of surface sulfates with catalytic activity. More detailed discussion needs to be added.

Figures 4 and 6 show the DRIFT spectra results comparing the two catalyst under NH3 and NO + O2 exposure, respectively. However, it is difficult to compare clearly that the adsorption capacity of NH3 and NO for the SSCO is improved by relative intensity comparisons in the y-axis range of different graphs.

This manuscript suggested that the performance of the catalyst is improved through sulfuric acid treatment, but the obvious results that can support this are only DRIFT and SCR activity. If the activity of the catalyst was improved by sulfation, it would be better to show how the catalytic activity varies (depending on the S content) according to the degree of acidification with sulfuric acid, rather than the two comparison groups in Figure 8.

In general, NH3-TPD analysis is carried out with In-Situ DRIFT analysis in order to determine Brønsted acid site and Lewis acid sites for SCR catalyst. As NH3 is adsorbed sufficiently and then the temperature is increased, desorption graph can be used to compare the acidic strength and the number of acid site. In this paper, the enhancement of catalytic acid sites by sulfuric acid treatment is the main content of the study, and it is believed that the addition of NH3-TPD results can more clearly support the improvement of catalytic activity.

XRD analysis showed that Fe2(SO4)3, FeSO4, and CaSO4 were detected in SSOC samples. Do these ingredients affect catalyst deactivation, such as pore plugging?

In the introduction section, the authors noted that the SCR is used as a representative NOx removal technology due to its low operation cost and high denitration efficiency. However, as you know, Vanadium-based SCR catalysts show a high NOx removal efficiency of over 90%. In the introduction, the authors need to add a additional description of why you have chosen a low-efficiency SOC catalyst.

Author Response

Question 1: In the DRIFT spectra results of Figure 3, the authors described that the two new peaks (965 and 927 cm-1) enhance the NH3 adsorption strength by sulfuric acid treatment and these sulfates on the surface of SSOC provide more Brønsted acid sites which can improve catalyst performance (p5, line 140). But there is a lack of explanation for the association of surface sulfates with catalytic activity. More detailed discussion needs to be added.

Answer: According to Eq 8, more Brønsted acid sites will result in more NO reduction. In addition, we investigated the acidity of catalyst on SCR performance, as shown in SFig 1. The higher acidity, the higher catalytic activity.

SFig. 1 The influence of acidity of SSOC on SCR performance

SSOC1 mean 100 g sintered ore sulfated by 0.05 mol sulpuric acid (1 mol/L sulpuric acid )

SSOC3 mean 100 g sintered ore sulfated by 0.15 mol sulpuric acid (3 mol/L sulpuric acid )

SSOC5 mean 100 g sintered ore sulfated by 0.25 mol sulpuric acid (5 mol/L sulpuric acid )

Question 2: Figures 4 and 6 show the DRIFT spectra results comparing the two catalyst under NH3 and NO + O2 exposure, respectively. However, it is difficult to compare clearly that the adsorption capacity of NH3 and NO for the SSCO is improved by relative intensity comparisons in the y-axis range of different graphs.

Answer: Thank you for your suggestions, we corrected the Fig.2 and Fig.6 according to your suggestion.

Question 3:This manuscript suggested that the performance of the catalyst is improved through sulfuric acid treatment, but the obvious results that can support this are only DRIFT and SCR activity. If the activity of the catalyst was improved by sulfation, it would be better to show how the catalytic activity varies (depending on the S content) according to the degree of acidification with sulfuric acid, rather than the two comparison groups in Figure 8.

Answer: I explained it in question 1. This part will be detail discussed in the next paper.

Question 4: In general, NH3-TPD analysis is carried out with In-Situ DRIFT analysis in order to determine Brønsted acid site and Lewis acid sites for SCR catalyst. As NH3 is adsorbed sufficiently and then the temperature is increased, desorption graph can be used to compare the acidic strength and the number of acid site. In this paper, the enhancement of catalytic acid sites by sulfuric acid treatment is the main content of the study, and it is believed that the addition of NH3-TPD results can more clearly support the improvement of catalytic activity.

Answer: Thank you for your good idea. We have a TPD, however, we can not use NH3 in this TPD at now. NH3 is corrosion gas and we need time to build a new apparatus.  

Question 5: XRD analysis showed that Fe2(SO4)3, FeSO4, and CaSO4 were detected in SSOC samples. Do these ingredients affect catalyst deactivation, such as pore plugging?

Answer: BET analysis demonstrated that the sulfation had a limit influence on the structure of the catalyst. The sulfation would block some pore, however, the sulfation also can open some micro-pore.

Question 6: In the introduction section, the authors noted that the SCR is used as a representative NOx removal technology due to its low operation cost and high denitration efficiency. However, as you know, Vanadium-based SCR catalysts show a high NOx removal efficiency of over 90%. In the introduction, the authors need to add an additional description of why you have chosen a low-efficiency SOC catalyst.

Answer: I corrected the introduction according to your suggestion in the revised version.

Reviewer 2 Report

This manuscript is well formatted and gives valuable information about sulfated sintered ore catalyst for NOx control. I think, this manuscript is acceptable for publication in the journal of Catalysts.

Author Response

Thank you for your review

Reviewer 3 Report

This paper deals with the DeNOx performance of sulfated sintered ore catalysts in comparison with original sintered ore catalysts. The results demonstrate a clear improvement of NOx elimination activity after SOC catalysts sulfation. I would recommend the publication of the manuscript after minor revision.

1) The authors should improve the introduction of the paper, including more references and explaining the improvement of SOC catalysts in comparison with the catalysts used nowadays for NOx removal in industry emissions.

2) Explain in Section 2.4 which are the gases measured at the outlet (at least NO, NO2, NH3 and N2O should be measured) and check if N-balance is closed under +-5%.

3) A deeper discussion must be included over Table 2, explaining in detail the differences and the reasons for BET surface area and pore volumes increase/decrease.

4) Fig. 1 must be improved, probably including bands representing each species. It seems difficult to determine which the species present are in (a) and (b).

5) DRIFT spectra results in Figure 3, need of deeper discussion. An explanation of the importance for DeNOx performance of the peaks at 960 and 930 must be given.

6) Also, Fig. 5 need of more discussion. All the peaks are crucial for NOx elimination? In line 160 it is stated that 300 ºC is the optimum temperature due to the maximum increase of every peak at this temperature. Please, clarify.

7) In Fig. 7 (a) and (b) must show same scale for comparison purposes, as explained in Lines 192-194.

8) Some phrases that must be rewritten or checked:

- Lines 30-31

- Line 81 (...two catalysts were...)

- Line 131 (...species, while the band...). Line 217 (GHSV, while the NOx conversion...)

- Check that all the units are separated from the number (lines 46, 215, 217).

Author Response

Question 1: The authors should improve the introduction of the paper, including more references and explaining the improvement of SOC catalysts in comparison with the catalysts used nowadays for NOx removal in industry emissions. 

Answer: I corrected the introduction according to your suggestion in the revised version.

Question 2:  Explain in Section 2.4 which are the gases measured at the outlet (at least NO, NO2, NH3 and N2O should be measured) and check if N-balance is closed under +-5%.

Answer: SFig.2 presented the spectra of the gas at inlet and outlet, the results showed there were mainly NO and NH3 with little NO2 in the gas at the inlet of reactor. After the SCR reaction, there was still limit amounts of N2O resulted from the side effects at the outlet. NH3 was not reacted completely, some ammonia remained at the outlet. The results showed that NO was mainly reduced to nitrogen. There were some side effects occurred in the reaction process, but the amount of the by-products was little. At this stage, we can not do N balance due to the accuracy of N2O in our analysis equipment.

Fig.2 DRFIT spectra of the gas at inlet and outlet

Question 3: A deeper discussion must be included over Table 2, explaining in detail the differences and the reasons for BET surface area and pore volumes increase/decrease.

Answer: Thank you for your suggestion, I added this part in the revised version.

Question 4: Fig. 1 must be improved, probably including bands representing each species. It seems difficult to determine which the species present are in (a) and (b).

Answer: Thank you for your suggestion, I corrected Fig 1 in the revised version.

Question 5:  DRIFT spectra results in Figure 3, need of deeper discussion. An explanation of the importance for DeNOx performance of the peaks at 960 and 930 must be given. 

Answer: I added the discussion in the revised version. These weakly adsorbed or gaseous NH3 could rapidly adsorbed on the acid sites once the adsorbed NH3 species was consumed, which promote SCR performance.

Question 6:  Also, Fig. 5 need of more discussion. All the peaks are crucial for NOx elimination? In line 160 it is stated that 300 ºC is the optimum temperature due to the maximum increase of every peak at this temperature. Please, clarify.

Answer: I added some discussion in the revised version. According to Eq 4-6, the more adsorbed bidentate nitrates and monodentate nitrates at the surface, the high catalytic activity.

Question 7: In Fig. 7 (a) and (b) must show same scale for comparison purposes, as explained in Lines 192-194.

Answer: I updated it in the revised version.

Question 8: Some phrases that must be rewritten or checked:

- Lines 30-31

- Line 81 (...two catalysts were...)

- Line 131 (...species, while the band...). Line 217 (GHSV, while the NOx conversion...)

- Check that all the units are separated from the number (lines 46, 215, 217).

Answer: Thank you for your suggestion, I corrected them in the revised version.

Round 2

Reviewer 1 Report

1. In the revised manuscript, the results of denitration performance of catalyst were moved to the 3. Experimental section (line 249). Because the NH3-SCR reaction is important results to reveal the catalytic activity, SCR performance results should be located after the 2.2 In-situ DRIFTS studies section.

2. The authors presented the Figure S1 showing the influence of acidity of synthesized SSOC on SCR performance in the response to the Reviewer’s comments. This is interesting results indicating the higher acidity, the higher catalytic activity. So, Figure S1 and following supplemental discussion should be added to the revised manuscript.

Author Response

Question 1: In the revised manuscript, the results of denitration performance of catalyst were moved to the 3. Experimental section (line 249). Because the NH3-SCR reaction is important results to reveal the catalytic activity, SCR performance results should be located after the 2.2 In-situ DRIFTS studies section.

Answer: I corrected it according to your suggestion, please see it in the revised version.

Question 2: The authors presented the Figure S1 showing the influence of acidity of synthesized SSOC on SCR performance in the response to the Reviewer’s comments. This is interesting results indicating the higher acidity, the higher catalytic activity. So, Figure S1 and following supplemental discussion should be added to the revised manuscript.

       Answer: I corrected it in the revised version, please see Fig. 5.